# Case Report: Identification of a Novel Variant (m.8909T>C) of Human Mitochondrial *ATP6* Gene and Its Functional Consequences on Yeast ATP Synthase

**DOI:** 10.3390/life10090215

**Published:** 2020-09-22

**Authors:** Qiuju Ding, Róża Kucharczyk, Weiwei Zhao, Alain Dautant, Shutian Xu, Katarzyna Niedzwiecka, Xin Su, Marie-France Giraud, Kewin Gombeau, Mingchao Zhang, Honglang Xie, Caihong Zeng, Marine Bouhier, Jean-Paul di Rago, Zhihong Liu, Déborah Tribouillard-Tanvier, Huimei Chen

**Affiliations:** 1National Clinical Research Center of Kidney Diseases, Jinling Hospital, Nanjing University School of Medicine, Nanjing 211166, China; dqj3206@163.com (Q.D.); 15205153118@163.com (W.Z.); tinne_xst@163.com (S.X.); zmchj99@163.com (M.Z.); xiehl_doctor@163.com (H.X.); zengch_nj@hotmail.com (C.Z.); liuzhihong@nju.edu.cn (Z.L.); 2Institute of Biochemistry and Biophysics, Polish Academy of Sciences, 00090 Warsaw, Poland; roza@ibb.waw.pl (R.K.); slimanio@gmail.com (K.N.); 3Institut de Biochimie et Génétique Cellulaires, Université de Bordeaux, CNRS, UMR 5095, F-33000 Bordeaux, France; a.dautant@ibgc.cnrs.fr (A.D.); xin.su@ibgc.cnrs.fr (X.S.); marie-france.giraud@ibgc.cnrs.fr (M.-F.G.); kewin.gombeau@ibgc.cnrs.fr (K.G.); marine.bouhier@ibgc.cnrs.fr (M.B.); jp.dirago@ibgc.cnrs.fr (J.-P.d.R.); 4Institut national de la santé et de la recherche médicale, 75000 Paris, France

**Keywords:** *MT-ATP6*, m.8909T>C, ATP synthase, nephropathy, oxidative phosphorylation, mitochondrial disease

## Abstract

With the advent of next generation sequencing, the list of mitochondrial DNA (mtDNA) mutations identified in patients rapidly and continuously expands. They are frequently found in a limited number of cases, sometimes a single individual (as with the case herein reported) and in heterogeneous genetic backgrounds (heteroplasmy), which makes it difficult to conclude about their pathogenicity and functional consequences. As an organism amenable to mitochondrial DNA manipulation, able to survive by fermentation to loss-of-function mtDNA mutations, and where heteroplasmy is unstable, *Saccharomyces cerevisiae* is an excellent model for investigating novel human mtDNA variants, in isolation and in a controlled genetic context. We herein report the identification of a novel variant in mitochondrial *ATP6* gene, m.8909T>C. It was found in combination with the well-known pathogenic m.3243A>G mutation in mt-tRNA^Leu^. We show that an equivalent of the m.8909T>C mutation compromises yeast adenosine tri-phosphate (ATP) synthase assembly/stability and reduces the rate of mitochondrial ATP synthesis by 20–30% compared to wild type yeast. Other previously reported *ATP6* mutations with a well-established pathogenicity (like m.8993T>C and m.9176T>C) were shown to have similar effects on yeast ATP synthase. It can be inferred that alone the m.8909T>C variant has the potential to compromise human health.

## 1. Introduction

Mitochondria provide aerobic eukaryotes with cellular energy by generating adenosine tri- phosphate (ATP) through the process of oxidative phosphorylation (OXPHOS). As a first step, electrons from nutrients (such as carbohydrates and fatty acids) are transferred by four complexes (CI-CIV) anchored to the mitochondrial inner membrane. This results in a transmembrane electrochemical proton gradient, which drives ATP synthesis from adenosine di-phosphate (ADP) and inorganic phosphate (Pi) by the ATP synthase (CV).

Defects of the OXPHOS system have been implicated in a broad spectrum of human diseases. Typical clinical traits include encephalopathies, cardiomyopathies, myopathies, visual/hearing, liver and renal dysfunctions [1,2,3]. Many of these diseases are caused by mitochondrial DNA (mtDNA) mutations. This DNA encodes 13 protein subunits of the OXPHOS system and a number of RNAs necessary for the synthesis of these proteins inside the organelle [4]. Pathogenic mtDNA mutations often co-exist with non-mutated mtDNA (heteroplasmy) and are highly recessive (only rare cases of dominancy were reported [5]), which makes it difficult to know how mitochondrial function is affected in patient’s cells and tissues. Furthermore, because of the high mutability of the mitochondrial genome due to its exposure to damaging oxygen species (ROS) and the poor activity of mitochondrial DNA repair systems, it can be difficult to establish the pathogenicity of a mtDNA variant [4]. Additionally, the effects of deleterious mtDNA mutations may be aggravated by nucleotide changes in nuclear and mitochondrial DNA that are not pathogenic per se (the so-called modifier genes) [6,7].

Being amenable to mitochondrial genetic transformation [8], and owing to its good fermenting capability that enables survival after the loss of oxidative phosphorylation and inability to stably maintain heteroplasmy [9], *Saccharomyces cerevisiae* has been used as a model to investigate mtDNA mutations identified in patients. We have exploited these attributes to study *MT-ATP6* gene mutations found in patients. This gene encodes the subunit *a* of ATP synthase, which is involved in moving protons through the membrane domain (F_O_) of ATP synthase coupled to ATP synthesis [10,11,12,13,14,15]. These yeast-based studies helped to better define the functional consequences of subunit *a* mutations and provided support for the pathogenicity of rare alleles identified in only a limited number of cases like m.8851T>C [12] and m.8969G>A [16]. Importantly, the extent to which yeast ATP synthase was affected correlated with disease severity, which reflects the strong evolutionary conservation of the regions of subunit *a* where these mutations localize [17,18,19,20,21,22,23,24,25,26].

Herein we report the identification of a novel *MT-ATP6* variant, m.8909T>C. It converts into serine a highly conserved phenylalanine residue of subunit *a* (*aF_128_S* in humans; *aF_145_S* in yeast). We detected it by entirely sequencing the mtDNA of a patient with an extreme clinical presentation including brain, kidney and muscular dysfunctions leading to premature death at the age of 14. This patient also carried the well-known pathogenic m.3243A>G mutation in mt-tRNA^Leu(1)^ (*MT-TL1*). In a yeast model of the m.8909T>C variant, ATP synthase assembly/stability was significantly compromised to an extent comparable to that seen previously in yeast models of other *ATP6* mutations with a well-established pathogenicity. These findings indicate that alone the m.8909T>C variant has the potential to compromise human health.

## 2. Materials and Methods

### 2.1. Kidney Analyses

Kidney biopsies were performed under ultrasound guidance by an experienced investigator. Paraffin-embedded sections were routinely stained with periodic acid Schiff and assessed by light microscopy. Fluorescence staining for IgG, IgA, immunoglobulin M (IgM), C3c and C1q was performed on freshly frozen renal tissues and ultra-thin sections stained with uranyl acetate and lead nitrate were examined by electron microscopy as described in [16]. Kidney biopsies were frozen in isopentane chilled with liquid nitrogen. Six-micrometer thick cryostat frozen sections were performed for enzyme histochemical staining for Complex IV (COX), Complex II (SDH) and nicotinamide adenine dinucleotide (NADH) dehydrogenase activities using specific substrates of these enzymes including Cytochrome *c* (C7752, Sigma-Aldrich), β-nicotinamide adenine dinucleotide (N7410, Sigma-Aldrich, St. Louis, MO, USA) and succinic acid (S3674, Sigma-Aldrich), respectively as described [27,28,29]. Optical density quantification of these activities was assessed from 20 consecutive microscopic fields in each renal section and the adjacent background regions.

### 2.2. Patient Consent, Ethical Committees, and Adhesion to Biosecurity and Institutional Safety Procedures

The female Chinese patient herein described was hospitalized with recurrent kidney disease and multiple systemic dysfunctions at Jinling Hospital in Nanjing (China). One hundred healthy control adults were randomly recruited from a panel of unaffected, genetically unrelated Han Chinese individuals from the same geographic region. All methods were conducted in accordance with the Ethics Committee of the Jinling Hospital and in the respect of biosafety and public health. Written informed consents were obtained from the patient and her parents as well as the 100 healthy controls. The mother didn’t consent for sequencing her mtDNA and medical examination.

### 2.3. mtDNA Characterization

Whole DNA was prepared from total blood, kidney and urine sediment samples with the DNA extraction kit from Qiagen (Hilden, Germany), and mtDNA was amplified as described in [16]. The revised Cambridge reference sequence (rCRS) of *H. sapiens* mitochondrial DNA (GenBank NC_012920.1) was used to identify variants of this DNA. Variant prioritization and the presence of deletion/depletion was performed as described [16]. The abundance of mutations was assessed using a Pyromark Q24 platform (Qiagen) as described in [16]. After purification using streptavidin Sepharose HP (GE Healthcare), and denaturation with NaOH, the PCR products were annealed and sequenced with the primers listed in Table 1.

### 2.4. Media for Growing Yeast

The following media were used for growing yeast. YPAD: 1% (*w*/*v*) yeast extract, 2% (*w*/*v*) bacto peptone, 40 mg/L adenine and 2% (*v*/*v*) glucose; YPAGly: 1% (*w*/*v*) yeast extract, 2% (*w*/*v*) bacto peptone, 40 mg/L adenine and 2% (*v*/*v*) glycerol; YPGALA: 1% (*w*/*v*) yeast extract, 2% (*w*/*v*) bacto peptone, 40 mg/L adenine and 2% (*v*/*v*) galactose. Solid media contained 2% (*w*/*v*) agar.

### 2.5. Construction of S. cerevisiae Strain RKY108 (aF_145_S)

We used the QuikChange XL Site-directed Mutagenesis Kit of Stratagene to introduce an equivalent of the m.8909T>C mutation (*aF_145_S*) in the yeast *ATP6* gene cloned in pUC19 [30], utilizing the oligonucleotide 5′-GGTTTATATAAACATGGTTGAGTATTCTTCTCATTATCAGTACCTGCTGGTACACCATTACC-3′. The *atp6*-F_145_S gene was cloned into the plasmid pJM2 [8]. The resulting plasmid (pRK66) was introduced into mitochondria of the strain DFS160 that totally lacks mitochondrial DNA (ρ^0^) as described [8] (see Table 2 for complete genotypes and sources of yeast strains). The resulting mitochondrial transformants (RKY109) were crossed to the *atp6*::*ARG8m* deletion strain MR10 [30] to yield strain RKY108. This strain has the MR10 nucleus and the *atp6*-F_145_S gene in a complete (ρ^+^) mitochondrial genome. The presence of the *atp6* mutation in these clones was confirmed by DNA sequencing. No other change was detected in the *ATP6* gene. The corresponding wild type strain MR6 (WT) is a derivative of strain W303-1B in which the nuclear *ARG8* gene has been replaced with *HIS3* (*arg8::HIS3*) and with the entirely sequenced mitochondrial genome of strain BY4741 [30].

### 2.6. Yeast-Based Drug Assay

0.05 OD_650nm_ of exponentially growing cells were homogeneously spread with sterile glass beads on a square Petri dish (12 cm × 12 cm) containing solid YPAGly medium. Sterile filters were deposited on the plate and spotted with oligomycin (purchased from Sigma, St. Louis, MO, USA) dissolved in DMSO.

### 2.7. Biochemical Investigation of Mitochondria

Oxygen consumption measurements in mitochondria isolated from yeast cells grown in complete galactose medium were performed according to [32], using a Clark electrode (Heito, Paris, France). Freshly prepared mitochondria were added to 1 mL of respiration buffer (10 mM Tris-maleate pH 6.8, 0.65 M sorbitol, 0.3 mM EGTA, and 3 mM potassium phosphate) at 0.15 mg/mL in the reaction chamber maintained at 28 °C. The oxygen consumption was measured after successive additions of 4 mM NADH, 150 µM ADP, 4 µM carbonyl cyanide m-chlorophenylhydrazone (CCCP). The rate of ATP synthesis was measured in the same conditions but with 750 µM ADP. Aliquots were withdrawn every 15 s from the reaction mix and supplemented with 3.5% (*w*/*v*) perchloric acid and 12.5 mM EDTA. After neutralization of the samples to pH 6.5 with KOH 0.3 M/MOPS, ATP was quantified by luciferin/luciferase assay (ATPLite kit from Perkin Elmer, Waltham, MA, USA) on a LKB bioluminometer. Oligomycin (3 μg/mL) was used to determine the amount of ATP produced by ATP synthase. ATPase activity was measured in non-osmotically protected mitochondria at pH 8.4 as described [33]. Blue native polyacrylamide gel electrophoresis (BN-PAGE) was performed as described [34]. For this, 200 µg of mitochondrial proteins were suspended in 50 µL of extraction buffer (30 mM HEPES, 150 mM potassium acetate, 12% glycerol, 2 mM 6-aminocaproic acid, 1 mM EGTA, 2% digitonin (Sigma), one protease inhibitor cocktail tablet (Roche) (pH 7.4) and incubated for 30 min on ice. After centrifugation (14,000 rpm, 4 °C, 30 min) the supernatant containing the solubilized complexes were supplemented with 2.25 µL of loading dye (5% Serva Blue G-250, 750 mM 6-aminocaproic acid) and run into NativePAGE 3–12% Bis-Tris Gel (Invitrogen, Carlsbad, CA, USA). After transfer onto PVDF membrane the yeast ATP synthase complexes were detected with polyclonal antibodies against α-F_1_ (Atp1), subunit *c* (Atp9) and subunit *a* (Atp6) used after 1:10,000, 1:5000, and 1:1000 dilutions respectively. The Atp1 antibodies were kindly provided by J. Velours. Anti-Atp9 antibodies were prepared by Eurogentec (Seraing, Belgium) with the synthetic peptide corresponding to the loop connecting the two transmembrane helices of Atp9 as an immunogen. The procedure used to in-gel visualize ATP synthase by its ATPase activity is described in [31].

### 2.8. Amino-Acid Alignments and Subunit a Topology

Clustal Omega [35] was used to compare amino acid sequences of subunits *a* of various species. The topology of the *aF_145_S* mutation was investigated according to the yeast ATP synthase structure described in [36] and drawn using PyMOL [37].

### 2.9. Statistical Analyses

Chi-square test and SPSS (16.0), Chicago, IL, USA. method were used for evaluating the statistical significance of the data. All of the tests were two tailed, and *p* values < 0.05 were considered significant.

## 3. Results

### 3.1. Case Report

The patient here reported suffered from hemiplegia, epileptic episodes, aphasia, blindness and deafness since she was 10 years old. Cranial CT (Computed Tomography) and MRI (Magnetic Resonance Imaging) revealed cerebral infarction and softening, and inflammatory changes. Moderate to severe abnormalities in electroencephalogram were observed. She also presented severe hearing impairment and hyper lactacidemia. At the age of 14 she developed a nephrotic syndrome with mass proteinuria, hypoproteinemia, and hyperlipidemia (Table 3). Renal biopsy indicated mild mesangial proliferative glomerulonephritis (MPGNs) pattern, with tubule atrophy and interstitial fibrosis (Figure 1A). Diffuse deposition of immunoglobulin M (IgM) was revealed by immunofluorescence microscopy (Figure 1B). Glomerular IgM deposition was observed in the patient’s kidney, but also to a lesser extent in healthy controls. IgM deposition can be caused by many reasons and was likely not a crucial factor in disease development. The patient’s neuromuscular manifestations partially mitigated with steroid and antiepileptic treatment. A supplemental ATP therapy and additional renal conservative treatment had also some benefits. The initial renal function impairment rapidly aggravated and the patient was at uremia stage when she died after 8 months of follow-up, as revealed by the levels of serum creatinine and urea nitrogen in blood (Table 3).

### 3.2. Hints for Mitochondrial Dysfunction

Electron microscopy revealed uneven and swelled mitochondria with barely detectable cristae in patient’s tubular epithelial cells (Figure 1C). Histochemical staining of cytochrome *c* oxidase (COX or CIV) and NADH dehydrogenase (CI) activities was strongly reduced in renal tissues from the patient in comparison to control kidney samples (Figure 1D), whereas the activity of succinate dehydrogenase (SDH) was much less diminished. No large rearrangement (deletion) and depletion of the patient’s mtDNA were detected (not shown). The mitochondrial genome of the patient was entirely sequenced, which revealed a number of nucleotide changes relative to the reference human mitochondrial genome (Table 4), among which m.3243A>G in mt-tRNA^Leu^, which is the most frequent pathogenic mtDNA allele [38] (Figure 2A). Another point mutation, never reported thus far, was detected in the *MT-ATP6*: m.8909T>C. It is absent in 2704 controls in databases and in 100 age-matched controls from the Nanjing geographic region from which the patient originated (Figure 2B). Pyrosequencing analyses revealed that the *MT-ATP6* variant was homoplasmic in blood, urine sediments (epithelial-like cells detached from tubules) and kidney, whereas the m.3243A>G change was heteroplasmic (50–90%). Defects in mitochondrial translation induced by the m.3243A>G mutation likely explain the poor histochemical staining of CI and CIV, two complexes of mixed genetic origin, and the much better preservation of the entirely nucleus-encoded SDH complex.

The m.8909T>C variant converts a phenylalanine residue into serine at position 128 of human subunit *a* (*a*F_128_S) of ATP synthase. This residue is highly conserved in a large panel of evolutionary distant species (see below). Therefore, if the m.3243A>G mutation in mt-tRNA^Leu^ likely impacted the two mtDNA encoded subunits (ATP6 and ATP8) of ATP synthase, as was observed in previous studies of cells containing this mutation [39,40,41], we considered that the m.8909T>C possibly affected also the ATP synthase. It would have been difficult to test this hypothesis from patient’s cells and tissues because of their mitochondrial genetic heterogeneity. We, therefore decided, as described below, to investigate the consequences in isolation of an equivalent of this mutation on the yeast ATP synthase.

### 3.3. Consequences of the m.8909T>C Mutation on Yeast ATP Synthase

Yeast subunit *a* (also called subunit *6* or Atp6) is synthesized as a precursor protein the first ten residues of which are removed during ATP synthase assembly [42]. The phenylalanine residue at position 128 of human subunit *a* that is changed into serine by the m.8909T>C mutation corresponds to the phenylalanine residue at position 145 in the mature yeast protein (155 in the unprocessed form) (see below). A yeast model homoplasmic for the m.8909T>C mutation was created by changing the phenylalanine codon TTC 155 into TCA (see Materials and Methods).

#### 3.3.1. Influence of the *aF_145_S* Mutation on Yeast Respiratory Growth

The *a*F_145_S mutant grew well from fermentable substrates like glucose (Figure 3A), where ATP synthase is not required. Mitochondrial-dependent growth on glycerol was also normal at both 28 °C (the optimal temperature for mitochondrial function in yeast), and at 36 °C (Figure 3A). However, in the presence of increasing concentrations of oligomycin, a chemical inhibitor of ATP synthase [43], respiratory growth of the *a*F_145_S mutant was less efficient compared to the WT (Figure 3A). An increased sensitivity to oligomycin is usually observed in yeast ATP synthase defective mutants because less of this drug is needed to reach the threshold of ATP synthase activity (20%) below which respiratory growth of yeast becomes obviously compromised [13,44]. The increased sensitivity to oligomycin of the *aF_145_S* mutant was further characterized by spreading the cells as a dense layer on glycerol medium and then exposed to a drop of oligomycin deposited on a sterile disk of paper (Figure 3B). Oligomycin diffuses in the growth medium, which results in the establishment of a continuous gradient around the disk. Growth is inhibited until a certain drug concentration. The halos of growth inhibition had a much higher diameter for the mutant vs. wild type yeast (Figure 3B), consistent with the growth tests shown in Figure 3A. These in vivo observations provide a strong indication that the *aF_145_S* mutation has detrimental consequences on ATP synthase.

#### 3.3.2. Influence of the *aF_145_S* Mutation on Mitochondrial Respiration and ATP Synthesis

The impact of the *aF_145_S* mutation on mitochondrial oxygen consumption and ATP synthesis was investigated in mitochondria extracted from cells grown at 36 °C in a rich galactose medium. Oxygen consumption was measured with NADH as an electron donor, alone (basal or State 4 respiration, which is induced only by the passive permeability to protons of the inner membrane), and after successive additions of ADP (State 3 or phosphorylating conditions, where respiration is normally twice stimulated vs. State 4) and the uncoupler CCCP (thus without any membrane potential, which further stimulates 2-fold the rate of respiration vs. State 3) (Table 5). State 4 respiration was not increased in the *aF_145_S* vs. *WT* mitochondria, indicating that the inner membrane had a normal passive permeability to protons and that there were no proton leak through the membrane domain (F_O_) of ATP synthase. At State 3 as well as in the presence of CCCP, the rate of oxygen consumption was decreased by 20–30% in the mutant vs. WT mitochondrial samples, and the rate of ATP synthesis (at State 3) was diminished in similar proportions (Table 5). These data indicate that the *aF_145_S* mutation slows down the rate of ATP production with no loss in the yield in ATP per electron transferred to oxygen. A decreased capacity to transfer electrons to oxygen is usually observed in yeast ATP synthase defective mutants, except in those with F_O_-mediated proton leaks [10,11,12,30,45], from which it was argued that the proton translocation activity of ATP synthase modulates biogenesis of the respiratory system presumably as a mean to co-regulate mitochondrial electron transfer and ATP synthesis activities [46].

Mitochondria were isolated from cell strains grown for 5–6 generations in YPGALA medium (rich galactose) at 36 °C. Reaction mixes contained 0.15 mg/mL of mitochondrial proteins, 4 mM NADH, 150 (for respiration assays) or 750 (for ATP synthesis) µM ADP, 4 µM CCCP, 3 µg/mL oligomycin (*oligo*). Respiratory and ATP synthesis activities were measured using freshly isolated, osmotically protected, mitochondria buffered at pH 6.8. The reported values are averages of two biological replicates and three technical replicates for each assay. Statistical significance of the data was tested using unpaired *t*-test (* indicates a *p*-value < 0.05).

#### 3.3.3. Influence of the *aF_145_S* Mutation on ATP Synthase Assembly/Stability

The ATP synthase organizes into a membrane-extrinsic domain (F_1_) and a domain (F_O_) largely anchored in the inner membrane [47,48,49]. The subunit *a* and a ring of identical subunits *c* move protons through the F_O_. As a result of this, the *c*-ring rotates and provokes conformational changes in the F_1_ that promote ATP synthesis. In current models, the assembly of ATP synthase starts with formation of F_1_, followed by its association to the *c*-ring and peripheral stalk subunits that prevent rotation of the catalytic subdomain (αβ_3_) of F_1_. The process ends with incorporation of subunit *a* [50]. When incorporation of subunit *a* is compromised, the F_1_ and the *c*-ring easily dissociate during BN-PAGE analysis of mitochondrial proteins, as was observed in a yeast strain lacking the *ATP6* gene [15,30,51]. In addition to fully assembled monomers and dimers of ATP synthase, free F_1_ and *c*-ring were detected in mitochondrial samples from the *aF_145_S* mutant whereas these particles were absent in those from *WT* yeast, as revealed by Western blot with subunit *c* (Atp9) antibodies and by the in-gel F_1_-mediated ATP hydrolytic activity (Figure 3C). The steady state levels of the mutated subunit *a* were estimated by Western blot (WB) with Atp6 antibodies of mitochondrial proteins resolved in denaturing gels. They were significantly reduced in the mutant vs. *WT* (Figure 3D). Taken together these data show that the *aF_145_S* change partially compromises a stable incorporation of subunit *a* within ATP synthase.

### 3.4. Topology of the Phenylalanine Residue Targeted by the m.8909T>C Mutation

Complete high-resolution structures of mitochondrial ATP synthase were described recently [36,47,48,49,52,53,54]. Near the middle of the membrane are located two universally conserved residues, an acidic one in subunit *c* (*c*E_59_ in yeast) and a positively charged arginine residue in subunit *a* (*a*R_176_ in yeast) that are functionally essential [36,55,56,57] (Figure 4B). Two hydrophilic clefts, one on the *p*-side and *n*-side of the membrane facilitate proton movement from one side of the membrane to the other. Inside the *p*-side channel, *c*E_59_ takes a proton from the intermembrane space (IMS) and releases it into the *n*-side cleft after an almost complete rotation of the *c*-ring. The phenylalanine residue 145 (128 in *H.s.*) that is changed into serine by the m.8909T>C mutation is located at the bottom of the *n*-side cleft within a cluster of hydrophobic residues (*aW_126_, aF_141_, aF_142_, aL_144_, aF_145_, aY_166_*) beneath the helical domain of subunit *a* (*a*H5) that runs along the subunit *c*-ring (Figure 4C). Replacement of *aF_145_* by a polar serine residue may weaken these hydrophobic interactions, and this is possibly responsible for the partial defects in ATP synthase assembly/stability observed in the *aF_145_S* mutant (as described above).

## 4. Discussion

With the advent of next generation sequencing methods, numerous novel variants of the mitochondrial DNA (mtDNA) are continuously identified in patients suffering from mitochondrial disorders. They are frequently found in only a limited number of cases, sometimes in a single individual (as with the case here reported), which makes it difficult to conclude about their pathogenicity. In a study aiming to probe the possible implication of mtDNA alterations in renal disease in the region of Nanjing, we examined more than 5000 patients with a biopsy-proven kidney dysfunction. Patients retained for entire mtDNA sequencing additionally showed at least two symptoms commonly observed in mitochondrial diseases like diabetes mellitus, deafness, neuromuscular and cardiac manifestations. Another criterion was the presence of abnormal mitochondrial ultrastructure and decreased cytochrome *c* oxidase (COX) and NADH (Complex I) dehydrogenase activities in kidney biopsies. In the patient with all these features here reported, we identified a novel variant in the mitochondrial *ATP6* gene, m8909T>C. It was homoplasmic in blood, urine sediments, and kidney (Figure 2B). It was detected in combination with the well-known pathogenic m.3243A>G mutation in a leucine tRNA gene (*MT-TL1)* [38], at a lesser abundancy (50–90%). There is no doubt that defects in mitochondrial translation induced by the m.3243A>G mutation largely contributed to the disease process and the severe decreases in CI and CIV activities, with only a minimal loss of SDH activity as was observed in other patients carrying this mutation [58].

The m.8909T>C variant leads to replacement of a well conserved phenylalanine residue with serine in ATP synthase subunit *a* (*aF_128_S*) (Figure 4A), which prompted us to investigate the consequences of an equivalent of this mutation (*aF_145_S*) in yeast. Several lines of evidence demonstrate a partial impairment of ATP synthase function in the mutant compared to wild type yeast: (i) respiratory growth showed a higher sensitivity to suboptimal concentrations of oligomycin, a chemical that inhibits ATP synthase; (ii) as was observed in many yeast ATP synthase defective mutants, the rate of mitochondrial oxygen consumption was diminished; (iii) ATP was produced less rapidly; and (iv) the presence in BN gels of partial ATP synthase assemblies (free F_1_ and *c*-ring particles) attested for a compromised ability of the mutated subunit *a* to be stably incorporated into ATP synthase (Figure 3). According to recently published atomic structures of yeast ATP synthase [36,54], these effects are presumably the consequence of a disorganization of a cluster of hydrophobic residues supposedly important to help subunit *a* to adopt a stable functional conformation around the *c*-ring. Similar ATP synthase defects were previously observed in yeast models of *MT-ATP6* mutations with a well-established pathogenicity, like m.9176T>C [13] and m.8993T>C [11]. It is thus a reasonable hypothesis that the m.8909T>C variant could by itself have detrimental consequences on human health.

In line with this hypothesis, the patient here described showed a very severe clinical presentation in comparison to 35 previously reported patients suffering from kidney problems and presumed to carry only the m.3243A>G mutation (in most of them the mtDNA was not entirely sequenced). The median onset age of disease was 26 years (Appendix A). More than 50% of these patients were first diagnosed with a renal disease due to persistent proteinuria. Glomerular lesions were observed in 21 patients, FSGS in 15 patients, and 3 cases showed TIN (Appendix A) [59,60,61,62,63,64,65,66,67,68,69,70,71,72,73,74,75,76,77,78,79,80,81]. At the time of diagnosis, two relatively aged patients (41 and 47 years old) developed ESRD. A total of 21 patients were recorded with a median follow-up period of 72 months (range 24–240 months). Kidney function remained stable in two patients after 24 and 72 months, respectively, and one patient died of heart failure 60 months after diagnosis. Eighteen patients evolved toward renal failure or ESRD (Appendix A) within 10-years of follow-up after diagnosis (Appendix A). Our patient, carrying both m.8909T>C and m.3243A>G, developed a kidney disease at the age of 14 and showed an extremely rapid progression to ESRD and death after only 8 months after diagnosis of a nephropathic syndrome.

As the m.8909T>C mutation was homoplasmic in the analyzed cells and tissues, it is likely that it is maternally inherited. Unfortunately, the patient’s mother did not consent to be sequenced and followed medically. She was apparently healthy at the time her daughter was admitted at the hospital. We had no more contact with her since the death of her child. From her apparent good health, at least at the age she brought her daughter at the hospital, we may conclude that the m.8909T>C mutation has no dramatic consequences on mitochondrial function, which is consistent with the relatively mild effects of an equivalent of this mutation on yeast ATP synthase. However, when combined to a more severe allele like m.3243G>A, it may have the potential to accelerate a disease process. Previous studies already concluded that well-known pathogenic *ATP6* variants can act in synergy with other genetic determinants in patients with a very severe clinical presentation. For instance, while alone it usually provokes mild clinical phenotypes [82], the m.9176T>C mutation was identified in a case of fulminant and fatal Leigh syndrome [23]. As there was no major difference in the amount of mutated mtDNA in relative’s probands, it was concluded that additional mitochondrial or nuclear genetic determinants were responsible for the different phenotypic expression. A second study similarly reported a rapid clinical evolution leading to sudden infant death syndrome (SIDS) in families segregating the m.8993T>G mutation [83] or the m.10044A>G mutation of the mt-tRNA^Gly^ gene [84], which suggested that mtDNA abnormalities should be considered as contributing to SIDS. Because of a lack of data from the mother of the patient here described, it would be premature to claim that m.8909T>C is a pathogenic mutation. However, considering the very severe clinical presentation of this patient, and the detrimental consequences of m.8909T>C on yeast ATP synthase, it is a reasonable proposal that this mutation has the potential to impact human health, in particular when combined to other genetic abnormalities that compromise mitochondrial function.

## 5. Conclusions

In conclusion, we here report a new mtDNA variant in the *ATP6* gene (m.8909T>C) and provide evidence using the yeast model that it has detrimental consequences on ATP synthase similar to those of other *ATP6* mutations with a well-established pathogenicity like m.9176T>C and m.8993T>C. On this basis, it is a reasonable assumption that this variant has the potential to compromise human health. Our work illustrates the power of yeast to help the study of specific human mtDNA variants found in heterogeneous genetic backgrounds and comprehension of human diseases linked to this DNA.

## Figures and Tables

**Figure 1 life-10-00215-f001:**
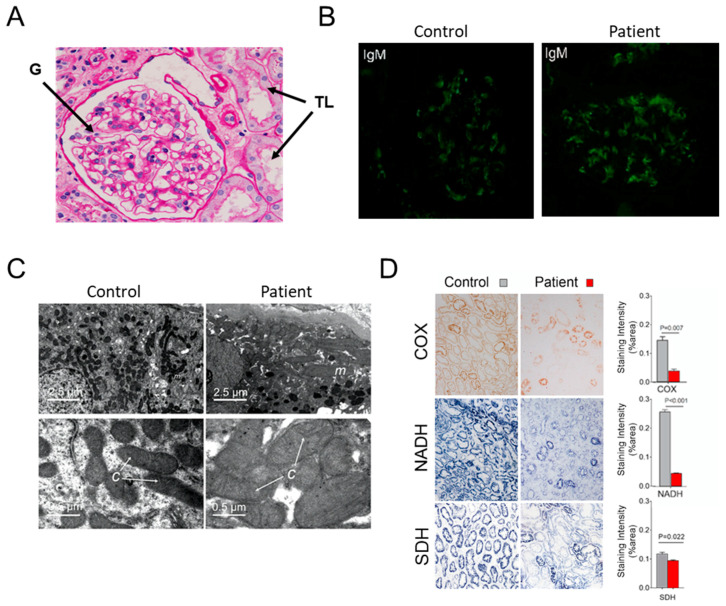
Kidney analyses. (**A**) Light microscopy of kidney samples from the patient show glomeruli (G) mesangial widening, and tubules (TL) atrophy and interstitial fibrosis. (**B**) Fluorescence microscopy reveals immunoglobulin M (IgM) deposits in glomerular cells from the patient. (**C**) Electron micrographs of kidney samples from the healthy control and patient (magnification is ×10,000, and ×80,000 from top to bottom). (**D**) Enzyme histochemical staining of cytochrome *c* oxidase (COX), nicotinamide adenine dinucleotide (NADH) and succinate dehydrogenase (SDH) in freshly frozen kidney biopsies (magnification is ×400).

**Figure 2 life-10-00215-f002:**
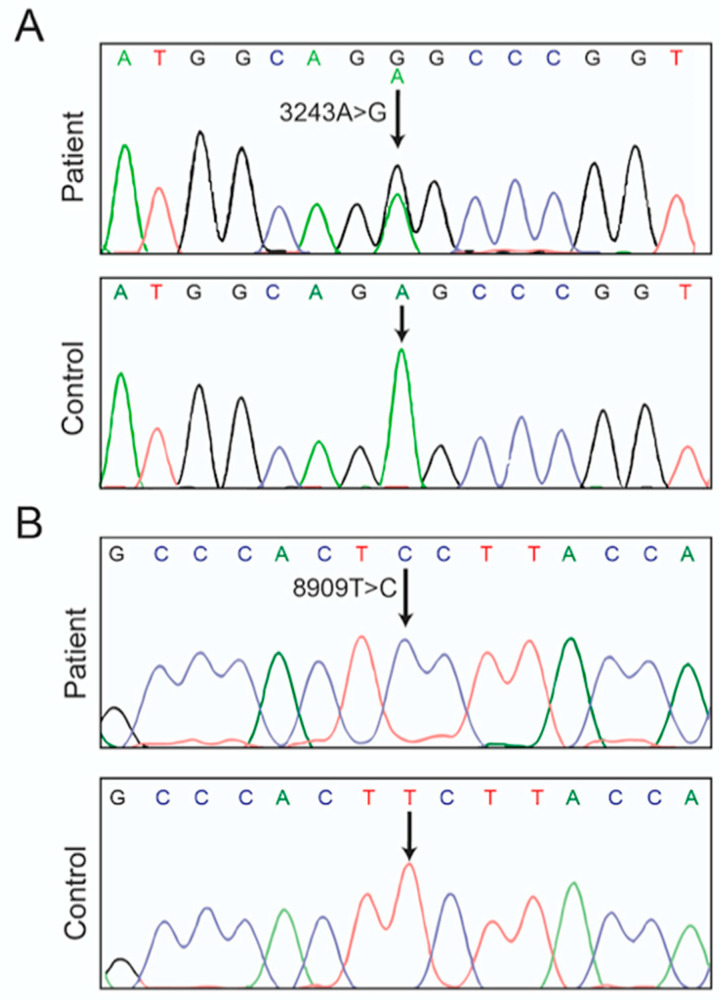
Mitochondrial DNA sequencing. Entire sequencing of the mtDNA of the patient identified the well-known pathogenic mutation m.3243A>G in *MT-TL1* (**A**) and a novel variant in *MT-ATP6*, m.8909T>C (**B**). The former was heteroplasmic whereas the latter was homoplasmic in analyzed cells and tissues (blood, urine and kidney). All the nucleotide changes relative to the reference human mitochondrial genome sequence are listed in Table 2.

**Figure 3 life-10-00215-f003:**
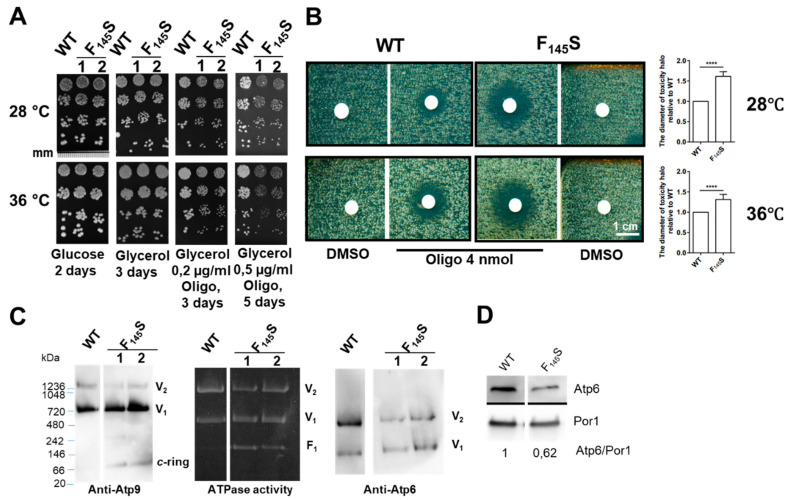
Consequences of an equivalent of the m.8909T>C variant (*aF_145_S*) in yeast. (**A**) Cells from the WT (MR6) and two genetically independent clones (denoted as 1 and 2) of the *aF_145_S* mutant strain (RKY108) grown in glucose were serially diluted and spotted on plates containing glucose, glycerol, and glycerol supplemented with indicated concentrations of oligomycin (Oligo). The plates were scanned after the indicated days of incubation. Representative data from two WT and two independent mutant clones and at least two repeats of each are shown. A mm scale is shown on the upper left image. (**B**) On the left panel, cells from WT and *aF_145_S* mutant strains were spread as dense layers onto rich glycerol solid media and then exposed to sterile filters spotted with 4 nmol oligomycin (Oligo) and DMSO as a negative control (solvent). The plates were scanned after 3 days of incubation at 28 °C and 36 °C. The shown scale is 1 cm. The diameters of the halos of growth inhibition (in % of WT) are reported in the shown histograms. **** indicates a *p-*value < 0.0001. (**C**) Assembly/stability of ATP synthase. Protein extracts were prepared from mitochondria isolated from WT and *aF_145_S* strains grown at 36 °C. Samples with a same content in porin were solubilized with 2% digitonin and resolved by Blue native–polyacrylamide gel electrophoresis (BN-PAGE, 200 µg proteins per lane). Dimers (V_2_) and monomers (V_1_) of F_1_F_O_ complexes and free F_1_ were in-gel visualized by their ATPase activity. The protein complexes were transferred onto nitrocellulose and probed with antibodies against subunit *c* (Atp9) and *a* (Atp6). On the left margin is a molecular weight ladder. (**D**) Western blot (WB) of mitochondrial proteins resolved by SDS-PAGE with antibodies against subunit *a* (Atp6) and Porin. The levels of Atp6 are normalized to Porin. The shown plates and gels have been cropped to eliminate samples not linked to this study and that were intercalated between those of interest (WT and *aF_145_S*). Representative data from at least two repeats are shown.

**Figure 4 life-10-00215-f004:**
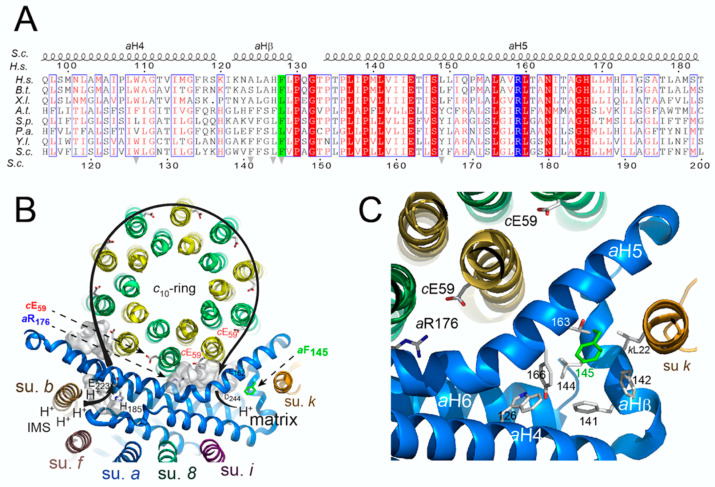
Evolutionary conservation of the phenylalanine residue targeted by the m.8909T>C mutation, and its topology in yeast ATP synthase. (**A**) Amino-acid alignments of subunits *a* from *Homo sapiens* (*H.s.*), *Bos taurus* (*B.t.*), *Xenopus laevis* (*X.l.*), *Arabidopsis thalian*a (*A.t.*), *Schizosaccharomyces pombe* (*S.p.*), *Podospora anser*ina (*P.a.*), *Yarrowia lipolytica* (*Y.l.*) and *Saccharomyces cerevisiae* (*S.c.*). At the top and bottom, the residues are numbered according to the *H.s.* protein and mature *S.c.* protein (i.e., without the first 10 N-terminal residues that are cleaved during assembly [42]), respectively. Strictly conserved residues are in white characters on a red background while similar residues are in red on a white background with blue frames. α-helices (*a*H4-5 and *a*Hβ) in the *S.c.* protein marked above the amino-acid alignments are according to [36]. The essential arginine (*a*R_159_ in humans, *a*R_176_ in yeast) is on a blue background. The yeast *aF_145_* residue corresponding to *a*F_128_ in *H.s.* targeted by the m.8909T>C mutation is on a green background. The grey arrows mark the residues belonging to the hydrophobic cluster surrounded by *a*H4, *a*Hβ and *a*H5. (**B**) View of the entire *c*-ring and subunits *a, b, 8, I, f and k* from the IMS and the pathway along which protons are transported from the intermembrane space to the mitochondrial matrix. The side chains of the residues that are essential (*a*R_176_ and *c*E_59_) and important (*a*E_162_, *a*D_244_, *a*E_223_, H_185_) to this transfer are drawn as stick with their carbon atoms in white. The *p*-side and *n*-side clefts are shown as grey surfaces. (**C**) Enlargement of the region where is located the *aF_145_S* mutation. The mutated *aF_145_* residue in green belongs to a hydrophobic cluster of six residues (*aW_126_, aF_141_, aF_142_, aL_144_, aF_145_, aY_166_*), which may stabilize or ease the folding of the bottom of the *n*-side cleft and the proper positioning of *a*H5 along the *c*-ring.

**Table 1 life-10-00215-t001:** Primers used for PCR amplification of mitochondrial DNA (mtDNA) and sequencing.

	Forward	Reverse	Product Length (bp)
For detection of point mutations			
1	CCCACAGTTTATGTAGCTTACC	GTACTATATCTATTGCGCCAGG	1215
2	ACTACCAGACAACCTTAGCC	AACATCGAGGTCGTAAACCC	1293
3	CTTCACCAGTCAAAGCGAAC	AGAAGTAGGGTCTTGGTGAC	1242
4	CGAACTAGTCTCAGGCTTCAAC	TCGTGGTGCTGGAGTTTAAG	1228
5	ACGTAAGCCTTCTCCTCACT	TCGTTACCTAGAAGGTTGCC	1138
6	CCGACCGTTGACTATTCTCT	GATGGCAAATACAGCTCCTA	1160
7	GCAAACTCATCACTAGACATCG	AGCTTTACAGTGGGCTCTAG	1329
8	ACCACAGTTTCATGCCCATC	TGGCCTTGGTATGTGCTTTC	1223
9	CACTTCCACTCCATAACGCT	GTTGAGGGTTATGAGAGTAGC	1308
10	TACCAAATGCCCCTCATTTA	GTAATGAGGATGTAAGCCCG	1272
11	TTCAATCAGCCACATAGCCC	GATGAAACCGATATCGCCGA	1260
12	GAGGGCGTAGGAATTATATCC	GTCAGGTTAGGTCTAGGAGG	1240
13	CATACTCGGATTCTACCCTAG	TGTAATTACTGTGGCCCCTC	1280
14	TCGGCATTATCCTCCTGCTT	GTGCTATGTACGGTAAATGGC	1250
15	TGACTCACCCATCAACAACC	ATAGAAAGGCTAGGACCAAACC	1179
For detection of mtDNA deletion			
1	GCACCCTATGTCGCAGTATCTGTCTTTG	GGACGAGAAGGGATTTGACTGTAATGTGC	16,255
2	CACTTCCACTCCATAACGCTCCTCATACT	GGGCTATTGGTTGAATGAGTAGGCTGATG	16,250
For detection of mtDNA copy number			
*COX1*	TTCGCCGACCGTTGACTATTCTCT	AAGATTATTACAAATGCATGGGC	197
*18S*	GTCTGTGATGCCCTTAGATG	AGCTTATGACCCGCACTTAC	177
For pyrosequencing of m.3243A>G			
Amplification	AAGGACAAGAGAAATAAGGC	ATGAGGAGTAGGAGGTTGG	207
Sequencing	TTTTATGCGATTACCG		

**Table 2 life-10-00215-t002:** Genotypes and sources of yeast strains.

Strain	Nuclear Genotype	mtDNA	Reference
DFS160	*MATα leu2Δ ura3-52 ade2-101 arg8::URA3 kar1-1*	ρ^o^	[31]
NB40-3C	*MATa lys2 leu2-3,112 ura3-52 his3ΔHindIII arg8::hisG*	ρ^+^ *cox2-62*	[31]
MR6	*MATa ade2-1 his3-11,15 trp1-1 leu2-3,112 ura3-1 CAN1 arg8::HIS3*	ρ^+^	[30]
MR10	*MATa ade2-1 his3-11,15 trp1-1 leu2-3,112 ura3-1 CAN1 arg8::HIS3*	ρ^+^ *atp6::ARG8^m^*	[30]
RKY109	*MATa leu2Δ* *ura3-52 ade2-101 arg8::URA3 kar1-1*	ρ*^−^ atp6-*F_145_S	This study
RKY108	*MATa ade2-1 his3-11,15 trp1-1 leu2-3,112 ura3-1 CAN1 arg8::HIS3*	ρ *atp6-*F_145_S	This study

Yeast genes nomencalure is used in accordance to http://www.yeastgenome.org/help/community/nomenclature-conventions.

**Table 3 life-10-00215-t003:** Clinical data.

The Time of First Kidney Biopsy	
Age/years	14
Sex (M/F)	F
Ethnicity	Han Chinese
Course/mouth	8
Urinary protein/g·24 h^−1^	3.74
BUN/mg·dL^−1^	20.6
Scr/mg·dL^−1^	1.23
Alb/g·L^−1^	27.5
Glo/g·L^−1^	16.8
TG/mmol·L^−1^	3.99
Chol/mmol·L^−1^	8.6
At uremia stage	
BUN/mg·dL^−1^	58.1
Scr/mg·dL^−1^	9.27

M: male; F: female; BUN: blood urea nitrogen; Scr: serum creatinine; Alb: albumin; Glo: globulin; TG: triglyceride; Chol: cholesterol.

**Table 4 life-10-00215-t004:** List of the mtDNA nucleotide changes in the patient relative to the reference sequence of the human mitochondrial genome (http://www.mtdb.igp.uu.se/).

Gene	Nucleotide Changes	Frequency ^a^	Amino Acid Change Function Annotation	Type
*D-Loop*	m.263A>G	1861/1867	non-coding	polymorphic
	m.499G>A *	2106/2144	non-coding	polymorphic
	m.16217T>C *	103/1867	non-coding	polymorphic
	m.16261C>T	111/1867	non-coding	polymorphic
	m.16136T>C *	24/1867	non-coding	polymorphic
*12S rRNA*	m.750A>G	2682/2704	non-coding	polymorphic
	m.827A>G *	54/2704	non-coding	polymorphic
	m.1438A>G	2620/2704	non-coding	polymorphic
*16S rRNA*	m.2706A>G	2178/2704	non-coding	polymorphic
*TL1*	m.3243A>G	0/2704	MELAS	Mutation
*ND2*	m.4769A>G	2674/2704	Met_100_Met	polymorphic
	m.4820G>A *	45/2704	Glu_117_Glu	polymorphic
	m.5063T>C	3/2704	Pro_198_Pro	polymorphic
*COXI*	m.6023G>A *	34/2704	Glu_40_Glu	polymorphic
	m.6413T>C *	23/2704	Asn_170_Asn	polymorphic
	m.7028C>T	2199/2704	Ala_375_Ala	polymorphic
*ATP6*	m.8860A>G	2698/2704	Thr_112_Ala	polymorphic
	m.8909T>C	0/2704	Phe128Ser	Mutation
*ND4L*	m.10646G>A	13/2704	Val_59_Val	polymorphic
*ND4*	m.11254T>C	1/2704	Ile_165_Ile	polymorphic
	m.11719G>A	2100/2704	Gly_320_Gly	polymorphic
*ND5*	m.13590G>A *	110/2704	Leu_418_Leu	polymorphic
	m.13779A>G	0/2704	Thr_481_Thr	polymorphic
*CYTB*	m.14766T>C	610/2704	Thr_7_Ile	polymorphic
	m.15326A>G	2687/2704	Thr_194_Ala	polymorphic
	m.15535C>T *	48/2704	Asn_263_Asn	polymorphic
	m.15688C>T	0/2704	Ser_314_Ser	polymorphic
	m.15758A>G	29/2704	Ile_338_Val	polymorphic

^a^ Frequency refers to the occurrence of the detected nucleotide changes in 2704 control individuals except D-Loop. * Stands for haplogroup B4b1a nucleotide changes.

**Table 5 life-10-00215-t005:** Mitochondrial respiration and ATP synthesis.

Strain	Respiration Rate nmoL O min^−1^ mg^−1^			ATP Synthesis Rate nmoL ATP min^−1^ mg^−1^	
	NADH	NADH + ADP	NADH + CCCP	−oligo	+oligo
*WT*	249 ± 21	654 ± 46	1028 ± 21	1393 ± 113	236 +/− 60
*aF_145_S*	178 ± 2 *	474 ± 5 *	811 ± 9 *	988 ± 101 *	110 +/− 49

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
