# Peer review of "Case Report: Identification of a Novel Variant (m.8909T>C) of Human Mitochondrial *ATP6* Gene and Its Functional Consequences on Yeast ATP Synthase"

_life, 2020, doi:10.3390/life10090215_

Round 1

Reviewer 1 Report

The authors report on a novel mtATP6 mutation m.8909T>C found in a patient carrying the well-known pathogenic mt-tRNAleu m.3243A>G mutation and exhibits a severe phenotype. To dissect the effects of both mutations the m.8909T>C variant was studied alone using a yeast model. They demonstrate that the equivalent of this mutation compromises the yeast ATP synthase assembly/stability and decrease the mitochondrial ATP synthesis rate. They conclude that the mtATP6 m.89091T>C mutation has the potential to compromise human health.

General remarks:

It is a nicely performed study providing interesting information’s. The hypothesis that not only quantitative differences in heteroplasmy of the mt-tRNAleu m.3243A>G mutation but also the occurrence of additional mtDNA mutations might explain the heterogeneity in the phenotypic expression related to the mt-tRNAleu m.3243A>G mutation is reasonable. However, to my mind the interpretation of the data as well as the conclusions are often over-estimated and partly speculative. The methods and statistics are partly poorly described. It is not comprehensible why they did not check the mtATP synthase properties in the patient during the 8 months observation period as a first check whether the ATP6 mutation might be biologically relevant in humans.

Specific comments

  1. Table 3 shows clinical data of the patient at a time point of first biopsy when kidney function was still normal as they stated in the text. Why later stages of renal disease were not included in the table? How many biopsies were taken during the 8 months? At line 69 p2, line 163 p6 and 369 p13 the authors state that the patient died at the age of 14, 8 months after first clinical signs of renal dysfunction occurred. In the table the age is given as 16 years at the time point of still normal renal function.

  1. Figure 1 – at which time point of the disease the kidney analysis was performed? The authors performed staining’s on frozen sections for COX, SDH and NADH dehydrogenase activities (statement line 196, 197) and measured the intensity of staining. Which method, which antibodies were used? It seems to me that they have done a normal immunohistochemistry – as such they measure the quantity of proteins not the activity. They should clearly differentiate between the terms “quantity” and “activity”. In figure 1D – quantification - how many controls were considered and how the quantification was performed (how many sections and how many areas per sections were analyzed). Why western blotting as a more reliable method for quantification was not used?

  1. Figure 3A - It is claimed that the suboptimal concentration of Oligomycin was sufficient to suppress growth in the mutant but not in the WT strain and concluded that the activity of the ATP synthase in mutants is less. Here, a series of oligomycin dilutions should be applied to proof that the threshold of effect is different between the WT and mutant cells. Moreover, I miss the quantification of the data. There seems to be considerable variations among WT probes. For instance at 28°C, the WT growth is considerable stronger in the group on glycerol with 0,2µg/ml Oligomycin, than in the group on glycerol without oligomycin. In my opinion this reflects the variability within the same treatment groups. That’s why it should be indicated how many samples and how many repeats/ sample were analysed. The data should be quantified.

Author Response

Reviewer 1

Comments and Suggestions for Authors

The authors report on a novel mtATP6 mutation m.8909T>C found in a patient carrying the well-known pathogenic mt-tRNAleu m.3243A>G mutation and exhibits a severe phenotype. To dissect the effects of both mutations the m.8909T>C variant was studied alone using a yeast model. They demonstrate that the equivalent of this mutation compromises the yeast ATP synthase assembly/stability and decrease the mitochondrial ATP synthesis rate. They conclude that the mtATP6 m.89091T>C mutation has the potential to compromise human health.

 General remarks:

It is a nicely performed study providing interesting information’s. The hypothesis that not only quantitative differences in heteroplasmy of the mt-tRNAleu m.3243A>G mutation but also the occurrence of additional mtDNA mutations might explain the heterogeneity in the phenotypic expression related to the mt-tRNAleu m.3243A>G mutation is reasonable.

We thank this reviewer for these positive comments

However, to my mind the interpretation of the data as well as the conclusions are often over-estimated and partly speculative. The methods and statistics are partly poorly described. It is not comprehensible why they did not check the mtATP synthase properties in the patient during the 8 months observation period as a first check whether the ATP6 mutation might be biologically relevant in humans.

The main contribution of this paper is the identification of a novel variant (m.8909T>C) of the human mitochondrial genome and we provide evidence that an equivalent of this mutation has detrimental consequences on yeast ATP synthase. Other ATP6 mutations (m.8993T>C, m.9176T>C) with a well-established pathogenicity showed similar effects in the yeast model [1, 2].  If, we agree that at this stage it is a speculative view that this variant contributed to the disease process in the patient where it was detected, and we took care not to claim that, we believe that it is a reasonable proposal that m.8909T>C has the potential to affect human health, in particular when combined to other genetic abnormalities (m.3243G>A in the present case) that compromise mitochondrial function, as was previously observed with m.9176T>C . 

Statistic treatment of data in Table 5 (lines 353-354) and quantification of growth tests (Fig. 3A,B see below) are provided.

Not surprisingly, because of their mixed genetic origin, Complexes I and IV showed a poor histochemical staining in patient’s tissues whereas the entirely nucleus-encoded Complex II was as expected much less affected (see also below). Not enough sample was made available to us to assess the content in Complex V. The Complex V was shown to be affected by the m.3243A>G mutation in previous studies [4-6], and this can make difficult to evaluate the consequences on this complex of the sole m.8909T>C mutation. Our investigation in isolation of an equivalent of this mutation in yeast provides good evidence that it has detrimental consequences on ATP synthase. We agree that testing the Complex V directly in patient’s tissues would have been an important source of information.

Specific comments

1. Table 3 shows clinical data of the patient at a time point of first biopsy when kidney function was still normal as they stated in the text. Why later stages of renal disease were not included in the table? How many biopsies were taken during the 8 months? At line 69 p2, line 163 p6 and 369 p13 the authors state that the patient died at the age of 14, 8 months after first clinical signs of renal dysfunction occurred. In the table the age is given as 16 years at the time point of still normal renal function.

The uremia stage of the patient after 8 months of follow-up has been included in Table 3. Electron and optic microscopy analyses (Fig. 1), performed when the patient developed at the age of 14 a nephrotic symptom, revealed typical renal alterations. No additional kidney biopsy was performed later. The patient died at the age of 14 after 8 months of follow-up, not at the age of 16 as was previously indicated by mistake in Table 3, which has been corrected. We have modified the text to indicate that kidney function was already impaired when the patient was taken in charge at the hospital (lines 218-220).

2. Figure 1 – at which time point of the disease the kidney analysis was performed? The authors performed staining’s on frozen sections for COX, SDH and NADH dehydrogenase activities (statement line 196, 197) and measured the intensity of staining. Which method, which antibodies were used? It seems to me that they have done a normal immunohistochemistry – as such they measure the quantity of proteins not the activity. They should clearly differentiate between the terms “quantity” and “activity”. In figure 1D – quantification - how many controls were considered and how the quantification was performed (how many sections and how many areas per sections were analyzed). Why western blotting as a more reliable method for quantification was not used?

Kidney analyses were performed at the time the patient was diagnosed with a nephrotic syndrome and admitted at the hospital. Enzymes responsible for oxidation of cytochrome c (COX), succinate (SDH) and NADH (Complex I) were investigated by normal histochemistry (without antibodies against these proteins). This assay reveals the activities and the location in tissues of enzymes that sustain them, as described in [7-9]. Briefly, tissue sections are incubated with a specific substrate, which is transformed into a primary reaction product that then combines with a capture agent forming a visible precipitate that co-localizes with the enzyme that sustains this activity. These assays are usually performed to evaluate the oxidative capacity of mitochondria in patients. A kidney slice from our patient was compared to two renal sections from healthy controls. Twenty consecutive microscopic fields in each renal section and the adjacent background regions were evaluated by optical density for semi-quantification of COX, SDH, and NADH activities. Technical information on these tests is provided (lines 115-122).

3. Figure 3A - It is claimed that the suboptimal concentration of Oligomycin was sufficient to suppress growth in the mutant but not in the WT strain and concluded that the activity of the ATP synthase in mutants is less. Here, a series of oligomycin dilutions should be applied to. Moreover, I miss the quantification of the data. There seems to be considerable variations among WT probes. For instance at 28°C, the WT growth is considerable stronger in the group on glycerol with 0,2µg/ml Oligomycin, than in the group on glycerol without oligomycin. In my opinion this reflects the variability within the same treatment groups. That’s why it should be indicated how many samples and how many repeats/ sample were analysed. The data should be quantified.

We fully agree with this comment. The tests have been repeated three times and the results obtained with two different concentrations of oligomycin are shown in the revised manuscript (Fig. 3A). Furthermore, for quantification, we used a second method in which the cells are spread as a dense layer on glycerol medium and then exposed to a drop of oligomycin deposited on a sterile disk of paper (Fig. 3B). Oligomycin diffuses in the growth medium, which results in a continuous gradient around the disk. Growth is inhibited until a certain drug concentration. The diameter of the resulting halo of growth inhibition allows a better comparison between strains. The two methods revealed a higher, statistically significant, sensitivity to oligomycin of the yeast model of m.8909T>C vs wild type yeast. In addition to incorporation of these new tests in Fig. 3, and commented in section 3.3.1 and a description of the second method is provided (lines 163-167).

Reviewer 2 Report

This paper by Ding, Kucharczyk, et al. identifies a novel mtDNA SNP and provides some functional evidence that it contributes to altered mitochondrial function. The data seems sound and the claims are not overly ambitious/interpreted. I think this paper will be a solid addition to the field and think it should be published as is.

Author Response

Comments and Suggestions for Authors

This paper by Ding, Kucharczyk, et al. identifies a novel mtDNA SNP and provides some functional evidence that it contributes to altered mitochondrial function. The data seems sound and the claims are not overly ambitious/interpreted. I think this paper will be a solid addition to the field and think it should be published as is.

We thank this reviewer very much for his/her positive comments

Reviewer 3 Report

I have four general comments on the manuscript. The first one concerns the presentation of the 3423 MELAS mutation by itself - for readers it really would be worthwhile to comment on a typical presentation without the additional mutation present, as the "exceptional severity" in the described patient is not compared to typical MELAS cases.

The second is something that is missing from the discussion - the mother's mtDNA was not sequenced, but if the patient has the 8909T>C mutation in the homoplasmic form and a heteroplastic MELAS mutation the mother probably was also heteroplasmic for that mutation but with a very high probability she was homoplasmic for the 8909T>C - was she healthy? She was at least healthy enough to have a child. Because if she was that mutation by itself is unlikely to be harmful in human cells, the authors could comment on this.

The third comment is that if the mtDNA was sequenced as it was the patient's haplogrup should be given, this is a sort of internal control that nothing was mixed up and that variants which are not found together do not occur together - there are a lot of papers by Salas and Bandelt on that subject.

and a minor thing - line 46 - most mtDNA mutations are indeed recessive but there is a dominant one (Sacconi et al., 2008 A functionally dominant mitochondrial DNA mutation)

And the last thing is the English - it is OK, but not perfect, I don't think I have listed everything, there's nothing serious there but it should be read carefully preferably by a native speaker (though the errors do not change the sense of the text, they are just a bit annoying)

line 27 - equivalent...compromise (should be compromises)

38 - are transferred made by complexes (why made?)

54 survival to the loss (should be after the loss)

160 10 years old (not year)

Table 3 - this is not just language, not race ethnic, but ethnicity

262-263 there was no proton leaks (should be were)

Author Response

Reviewer 3

Comments and Suggestions for Authors

I have four general comments on the manuscript. The first one concerns the presentation of the 3423 MELAS mutation by itself - for readers it really would be worthwhile to comment on a typical presentation without the additional mutation present, as the "exceptional severity" in the described patient is not compared to typical MELAS cases.

This is important indeed. To appreciate the severity of the clinical presentation of our patient we reviewed 35 previously reported m.3243G>A cases suffering from kidney problems (Tables S1 and S2, and Figure S1). The median onset age of disease was 26 years. >50% of these patients were first diagnosed with a renal disease due to persistent proteinuria. Glomerular lesions were observed in 21 patients, FSGS in 15 patients, and 3 cases had TIN. At the time of diagnosis, 2 relatively aged patients (41 and 47 years old) developed ESRD, but detailed follow-up information was not reported. 21 patients were recorded with a median follow-up period of 72 months (range 24-240 months). Kidney function remained stable in two patients after 24 and 72 months, respectively, and one patient died of heart failure 60 months after diagnosis. Eighteen patients suffered from renal failure or ESRD, and the cumulative 2-, 5- and 10-year overall renal failure rates were 4.76%, 29.83% and 77.9%, respectively (Figure S1). Our patient, carrying both m.8909T>C and m.3243A>G, developed a kidney disease at the age of 14, which is relatively early comparing with single m.3243G>A mutation, and she showed an extremely rapid progression to ESRD and death after only 8 months after diagnosis. On this basis, it seems to us correct to say that our patient had a particularly severe presentation compared to typical MELAS cases. In addition to the new Tables S1 and S2, and Figure S1, we have accordingly modified the text (lines 428-440).

The second is something that is missing from the discussion - the mother's mtDNA was not sequenced, but if the patient has the 8909T>C mutation in the homoplasmic form and a heteroplastic MELAS mutation the mother probably was also heteroplasmic for that mutation but with a very high probability she was homoplasmic for the 8909T>C - was she healthy? She was at least healthy enough to have a child. Because if she was that mutation by itself is unlikely to be harmful in human cells, the authors could comment on this.

This is an important point indeed. Unfortunately, the patient’s mother did not consent to be sequenced and followed medically. She was apparently healthy at the time her daughter was diagnosed with a nephrotic syndrome and admitted at the hospital. We have not had any more contact with her since the death of her child. As pointed by this reviewer, the mother likely harbors the m.8909T>C variant. From her apparent good health, at least at the age she brought her daughter at the hospital, we may indeed conclude that the m.8909T>C mutation has no dramatic consequences on mitochondrial function, which is consistent with the relatively mild effects of an equivalent of this mutation on yeast ATP synthase. However, when combined to a more severe allele like m.3243G>A, it may have the potential to accelerate a disease process. This was previously reported with the m.9176T>C mutation as a contributing factor leading to a fulminant Leigh syndrome in combination with other genetic abnormalities, while alone it induces relatively mild clinical phenotypes [3]. We have commented this important issue in the Discussion (lines 441-449).

The third comment is that if the mtDNA was sequenced as it was the patient's haplogrup should be given, this is a sort of internal control that nothing was mixed up and that variants which are not found together do not occur together - there are a lot of papers by Salas and Bandelt on that subject.

Sequence analysis of mitochondrial genome in this patient identified 10 variants belonging to Asian haplogroup B4b1a, which has been specified in Table 4.

and a minor thing - line 46 - most mtDNA mutations are indeed recessive but there is a dominant one (Sacconi et al., 2008 A functionally dominant mitochondrial DNA mutation).

Thank you for bringing to our attention this interesting case, which is now quoted in the revised manuscript (line 82, Ref N°5).

And the last thing is the English - it is OK, but not perfect, I don't think I have listed everything, there's nothing serious there but it should be read carefully preferably by a native speaker (though the errors do not change the sense of the text, they are just a bit annoying)

Line 27 - equivalent...compromise (should be compromises)

This has been corrected.

Line 38 - are transferred made by complexes (why made?)

This has been corrected (line 73).

Line 54 survival to the loss (should be after the loss)

This has been corrected (line 90)

Line 160 10 years old (not year)

This has been corrected (line 208).

Table 3 - this is not just language, not race ethnic, but ethnicity

This has been corrected (Table 3).

lines 262-263 there was no proton leaks (should be were)

has been corrected (line 306).

  1. Kucharczyk, R., M. Rak, and J.P. di Rago, Biochemical consequences in yeast of the human mitochondrial DNA 8993T>C mutation in the ATPase6 gene found in NARP/MILS patients. Biochim Biophys Acta, 2009. 1793(5): p. 817-24.
  2. Kucharczyk, R., et al., Consequences of the pathogenic T9176C mutation of human mitochondrial DNA on yeast mitochondrial ATP synthase. Biochim Biophys Acta, 2010. 1797(6-7): p. 1105-1112.
  3. Dionisi-Vici, C., et al., Fulminant Leigh syndrome and sudden unexpected death in a family with the T9176C mutation of the mitochondrial ATPase 6 gene. J Inherit Metab Dis, 1998. 21(1): p. 2-8.
  4. Fornuskova, D., et al., The impact of mitochondrial tRNA mutations on the amount of ATP synthase differs in the brain compared to other tissues. Biochim Biophys Acta, 2008. 1782(5): p. 317-25.
  5. Sasarman, F., H. Antonicka, and E.A. Shoubridge, The A3243G tRNALeu(UUR) MELAS mutation causes amino acid misincorporation and a combined respiratory chain assembly defect partially suppressed by overexpression of EFTu and EFG2. Hum Mol Genet, 2008. 17(23): p. 3697-707.
  6. McMillan, R.P., et al., Quantitative Variation in m.3243A > G Mutation Produce Discrete Changes in Energy Metabolism. Sci Rep, 2019. 9(1): p. 5752.
  7. Whitaker-Menezes, D., et al., Hyperactivation of oxidative mitochondrial metabolism in epithelial cancer cells in situ: visualizing the therapeutic effects of metformin in tumor tissue. Cell Cycle, 2011. 10(23): p. 4047-64.
  8. Kiernan, J.A., Indigogenic substrates for detection and localization of enzymes. Biotech Histochem, 2007. 82(2): p. 73-103.
  9. Hench, J., et al., A tissue-specific approach to the analysis of metabolic changes in Caenorhabditis elegans. PLoS One, 2011. 6(12): p. e28417.

Reviewer 4 Report

The article “Case report: Identification of a novel variant of human mitochondrial

ATP6 gene and its functional consequences on yeast ATP synthase” is well written.

It does not contain any grammatical mistakes.

The aim of the study is rather clear.
The work has a high degree of novelty.

The manuscript can be recommended for publication in the journal after minor revision.

It is recommended for Table 1 to place the headline near the very table.

It is recommended to insert a list of abbreviations into the article.

It is recommended to include chapter “Сonclusion” into the article.

It is recommended to add links to articles of 2018-2020 in chapter "References".

Author Response

Reviewer 4

Comments and Suggestions for Authors

The article “Case report: Identification of a novel variant of human mitochondrial

ATP6 gene and its functional consequences on yeast ATP synthase” is well written.

It does not contain any grammatical mistakes.

The aim of the study is rather clear.

The work has a high degree of novelty.

The manuscript can be recommended for publication in the journal after minor revision.

We thank this reviewer very much for his/her positive comments.

It is recommended for Table 1 to place the headline near the very table.

We are sorry, we failed to this despite many assays and we had problems with other Tables. Hopefully Life will arrange this.

It is recommended to insert a list of abbreviations into the article.

This has been done (lines 35-68).

It is recommended to include chapter “Сonclusion” into the article.

This has been done (lines 466-474).

It is recommended to add links to articles of 2018-2020 in chapter "References".

We apologize for not understanding this recommendation.

Round 2

Reviewer 1 Report

The authors adequately responded to the comments and provided all necessary informations

This manuscript is a resubmission of an earlier submission. The following is a list of the peer review reports and author responses from that submission.

Round 1

Reviewer 1 Report

Comments to the authors:

Qiuju Ding et identified a novel variant of human ATP6 gene, m.8909T>C, in a yung female patient. The authors demonstrated that, in a yeast model of the m.8909T>C variant, the stability of the ATP synthase was significantly compromised. In this work the authors hypothesize that this novel variant had detrimental consequences on human health.

The paper is well written however, I have questions and suggestions to the authors:

. Please change the abstract section. I suggest to introduce the topic of the study before results.

. Please insert the urinary and blood parameters of the patient. The authors indicated that the patient developed nephrotic syndrome with proteinuria, hypoproteinemia ad hyperlipidemia. What was her renal function? It is important to understand the renal demage.

. I suggest to specify in the title that it is a case report.

Author Response

Comments to the authors:

Qiuju Ding et identified a novel variant of human ATP6 gene, m.8909T>C, in a yung female patient. The authors demonstrated that, in a yeast model of the m.8909T>C variant, the stability of the ATP synthase was significantly compromised. In this work the authors hypothesize that this novel variant had detrimental consequences on human health.

The paper is well written however, I have questions and suggestions to the authors:

. Please change the abstract section. I suggest to introduce the topic of the study before results.

Thank you for this suggestion. As a result, quite important modifications were introduced and to stay within the 200 words limit we have removed some details present in the original version (lines 17 to 31). It seems to us that these changes in abstract give a much better visibility and comprehension of the scope of our work.

. Please insert the urinary and blood parameters of the patient. The authors indicated that the patient developed nephrotic syndrome with proteinuria, hypoproteinemia ad hyperlipidemia. What was her renal function? It is important to understand the renal demage.

The urinary and blood parameters of the patient have been added (novel Table 3, line 269). Renal function is here according to the values of serum creatinine and urea nitrogen in blood. At the time of first kidney biopsy, renal function was normal in the patient (Scr = 1.23 mg·dl-1; BUN = 20.6 mg·dl-1). It then deteriorated rapidly (Scr = 9.27 mg·dl-1 and BUN = 58.1 mg·dl-1) and the patient was in the period of uremia stage when she died after 8 months of follow-up. The text has been modified accordingly (lines 248-251).

. I suggest to specify in the title that it is a case report.

This has been done (line 2).

Reviewer 2 Report

Ding and coworkers described a patients with mitochondrial disease caused by an already reported  pathogenic variant in mttRNA Leu1 and an a new variant in the ATP6 gene.

The patients exhibited multiple OXPHOS deficiency characterized by CI and CIV defect in kidney biopsies that can be explained by the pathogenic variant in mttRNA Leu1.  Next the author focused on demonstrating the pathogenicity of the variant in ATP6 using yeast as a model, without first performing the most logic and easy investigation for establishing genetic diagnosis: mtDNA sequencing of mother and grand mother? CV defect in kidney biopsies?

Major points:

1.While I understand the value of the yeast model in this particular case, where mutation in mttRNA Leu1 could also leads to CV deficiency, I wonder why the authors did not perform BNPAGE analysis of CV and  western blot for the ATP6 protein level in kidney biopsies. In case of no CV defect we could already conclude that ATP6 variation is not pathogenic. In the case of CV defect, the studies in yeast are justified.

2.Moreover the authors presented histochemistry data concerning CI and CIV activity. Why the authors did not performed also the SDH activity? CII activity should not be affected in the case of mitochondrial translation defect, but it represents a good control to test the quality of control and patient biopsies.

3.Were the variants in mttRNA Leu1 and ATP6 present also in the mother, grand-mother and siblings? This is a really important piece of information to understand the pathogenicity of the variant in ATP6, since it is homoplasmic. What do you mean with “no available familial history”?

4.Figure 3 presents yeast growth, BNPAGE and in gel activity of control and yeast carrying the ATP6 variant. Because all the figures are cropped between control and mutant yeast, can you provide the original figures to verify that they are coming from the same gel/ plate?

5.Please can you verify the expression of the wt ATP6 and mtATP6 in the yeast? Is the mutation destabilizing the ATP6 protein level, as it is the case for the majority of the missense mutation?

Minor points:

1.no information or reference for the in gel activity experiment

2.figure 1D legend: “in fresh kidney biopsies”….but in mat and met biopsies were frozen. Please clarify.

3.there is no explanation on why the deposition of IgM in kidney biopsies is relevant. There is no control kidney for the investigation of IgM.

Author Response

Reviewer 2

Comments and Suggestions for Authors

Ding and coworkers described a patients with mitochondrial disease caused by an already reported  pathogenic variant in mttRNA Leu1 and an a new variant in the ATP6 gene.

The patients exhibited multiple OXPHOS deficiency characterized by CI and CIV defect in kidney biopsies that can be explained by the pathogenic variant in mttRNA Leu1.  Next the author focused on demonstrating the pathogenicity of the variant in ATP6 using yeast as a model, without first performing the most logic and easy investigation for establishing genetic diagnosis: mtDNA sequencing of mother and grand mother? CV defect in kidney biopsies?

Major points:

1.While I understand the value of the yeast model in this particular case, where mutation in mttRNA Leu1 could also leads to CV deficiency, I wonder why the authors did not perform BNPAGE analysis of CV and  western blot for the ATP6 protein level in kidney biopsies. In case of no CV defect we could already conclude that ATP6 variation is not pathogenic. In the case of CV defect, the studies in yeast are justified.

Because of its relatively high abundance, it is expected that alone the m.3243A>G mutation in mt-tRNALeu will already strongly impact the two mtDNA encoded subunits of Complex V. As a consequence, it might be difficult by probing the CV in the patient’s kidney to know to which extent this complex is affected by the m.8909T>C mutation alone. This is the reason why we investigated the consequences in isolation of m.8909T>C on ATP synthase in yeast. However, we thank this reviewer for this interesting comment and have accordingly added a few words on this point in the manuscript (line 291-320) to further justify the choice and utility of the yeast model to gain insight into m.8909T>C mutation because of the patient’s mitochondrial genetic heterogeneity.

2.Moreover the authors presented histochemistry data concerning CI and CIV activity. Why the authors did not performed also the SDH activity? CII activity should not be affected in the case of mitochondrial translation defect, but it represents a good control to test the quality of control and patient biopsies.

Indeed, the CII should not be directly affected by the m.3243A>G-induced mitochondrial translational defect. However, this complex might be indirectly impacted because of major defects in supramolecular organization of RC complexes (respirasome) as was observed with specific RC defects (in CI for instance). Furthermore, beyond a lack in mt-DNA encoded proteins, translational defects may have additional deleterious consequences due to the overaccumulation of aberrant and toxic mitochondrial translation products that the ATP-consuming mitochondrial protein quality control systems will fail to properly eliminate. We thus expect that a decreased SDH activity would be observed too in patient’s tissues, which would not be necessarily the result of some nuclear genetic abnormality.

3.Were the variants in mttRNA Leu1 and ATP6 present also in the mother, grand-mother and siblings? This is a really important piece of information to understand the pathogenicity of the variant in ATP6, since it is homoplasmic. What do you mean with “no available familial history”?

This reviewer is fully right that this is an important missing piece of information. This is not because we didn’t try to get it. As discretely indicated in the Materials and methods section of the original paper, the patient’s mother didn’t consent to be investigated and her mtDNA sequenced, and there was no other available proband’s relative from whom this information could have been be obtained (which we have referred to by “no available familial history”, which we agree is a bit vague).  As a result, we took care not to claim that the m.8909T>C is pathogenic, everywhere in the manuscript. Instead, because (i) it was novel (not mentioned in any database); (ii) combined with another mutation (m.3243A>G) with a well-established pathogenicity making it difficult to assess its functional consequences from patient’s cells and tissues; and (iii) because it affects a highly conserved region of subunit a presumably important from high-resolution structural data for properly shaping/stabilizing this protein, we have limited our conclusions mainly to the impact of this mutation in isolation  on ATP synthase using the yeast model. Since an equivalent of the m.8909T>C had on the yeast ATP synthase effects similar to those of other ATP6 mutations with a well-established pathogenicity (e.g. m.8993C, m.9176T>C), we believe that our work provides an interesting piece of information from which it can be concluded that the m.8909T>C has by its own the potential to compromise human health. Would this mutation be identified in the future in other individuals, our study may be an important source of information for the comprehension of their health problems. Furthermore, our work provides the first illustration of how the yeast system can help to study specific human mtDNA mutations found in heterogenous genetic backgrounds where several mutation of this DNA co-exist, which is a useful approach to better understand human diseases linked to mtDNA.

We have accordingly modified the manuscript to better underline this key point raised by this reviewer and have tried to rephrase in a more understandable way the notion of “no available familial history” (lines 538, 570-573, 587-590).

4.Figure 3 presents yeast growth, BNPAGE and in gel activity of control and yeast carrying the ATP6 variant. Because all the figures are cropped between control and mutant yeast, can you provide the original figures to verify that they are coming from the same gel/ plate?

We fully understand this request. The cropping is because there was some samples (from another mutant (S175N)) not linked to the present study intercalated between the WT and the F145S yeast strains (see below). Judging that it may not be appropriate to show the whole plates and gels, we have indicated in the legends of Fig.3, that the data are coming from the same gels/plates (lines 480-482). Would you consider that these documents should not be cropped and shown entirely, we would accept to do it.

Non-cropped Figure 3A  (see the word file)

Non-cropped Figure 3B (see the word file)

5.Please can you verify the expression of the wt ATP6 and mtATP6 in the yeast? Is the mutation destabilizing the ATP6 protein level, as it is the case for the majority of the missense mutation?

We had made but not included this test in the original manuscript that we agree is an interesting information and have therefore included it. In accordance with the BN-PAGE data. the shown WB reveals that the steady state levels of Atp6/subunit is significantly diminished in the F145S mutant compared to wild type yeast. This result is mentioned in the manuscript (lines 462-465) and we have added a new panel in Fig.3 (panel C) showing this. As for Figs. 3A and 3B, the shown WB is cropped to remove the sample (S175N) not linked to this study. Here is the non-cropped figure 3C:

 Non-cropped Figure 3C (see the word file)

We have also added WB with Atp6 antibodies (in Fig. 2B) of proteins resolved by BN-PAGE showing a significant decrease in the content of assembled ATP synthase in the mutant vs WT.

Minor points:

1.no information or reference for the in gel activity experiment

The reference (n°31) describing this procedure is now provided (lines 225-226).

2.figure 1D legend: “in fresh kidney biopsies”….but in mat and met biopsies were frozen. Please clarify.

We apologize for the confusion. These analyses were made with freshly frozen renal tissues. We have modified accordingly the legend of Fig. 1D (line 266) and Materials and methods (line 133).

3.there is no explanation on why the deposition of IgM in kidney biopsies is relevant. There is no control kidney for the investigation of IgM.

We have added a control kidney for the investigation of IgM in Figure 1B. IgM deposition was observed in the control but at a lesser extent compared to the patient. Glomerular IgM deposits can be caused by many reasons. It was likely not a crucial factor in the development of the disease we described. We have accordingly modified the text (lines 242-246).

  Figure (see the word file)

Round 2

Reviewer 1 Report

The authors have provided a new version of the manuscript answering to the referees' comments and suggenstions.

In this form i can accept this work.

Reviewer 2 Report

After reading the answer of the authors,  I do not agree with several of their arguments:

  1. As I stated before, I understand the reason why the authors used yeast as a model to investigate the variant in ATP6. But to my question concerning CV activity/biogenesis in kidney biopsies remains still without answer. The authors says : it is expected that alone the m.3243A>G mutation in mt-tRNALeu will already strongly impact the two mtDNA encoded subunits of Complex V. But in science nothing is expected, everything needs to be verify experimentally and in this particular patient is extremely important to know if there is CV defect.

  1. I asked to use histochemistry assay for CII activity in kidney biopsies because CII is the only OXPHOS complex with nuclear encoded subunits and it is not affected directly by mutation m.3243A>G beside being a good control for quality preservation of the biopsies. The authors reply with a weird argument that CII might be indirectly impacted because of major defects in supramolecular organization of RC complexes (respirasome). First of all CII is the only OXPHOS complex that is not part of the respirasome and moreover it is well recognize that in patients with isolated or multiple OXPHOS activities, CII is not affected, but on the contrary, its steady state level is sometimes increased as part of compensatory mechanisms. Performing SDH staining (CII activity) in the biopsies is the only way to know if the biopsies were stored properly and  this will strengthen the results concerning CI and CIV defect.

  1. Thank you for providing the original files of the pictures. Clearly the wt of Figure 3B (CV in gel activity) is not the same control of the original picture. Comments?                                                                  Please make visible that the figures were cropped.